# Co-Translational Quality Control Induced by Translational Arrest

**DOI:** 10.3390/biom13020317

**Published:** 2023-02-07

**Authors:** Yoshitaka Matsuo, Toshifumi Inada

**Affiliations:** Division of RNA and Gene regulation, Institute of Medical Science, The University of Tokyo, Tokyo 108-8639, Japan

**Keywords:** quality control, transrational arrest, ribosome collision, ubiquitination, protein degradation, mRNA decay, non-canonical ribosome dissociation

## Abstract

Genetic mutations, mRNA processing errors, and lack of availability of charged tRNAs sometimes slow down or completely stall translating ribosomes. Since an incomplete nascent chain derived from stalled ribosomes may function anomalously, such as by forming toxic aggregates, surveillance systems monitor every step of translation and dispose of such products to prevent their accumulation. Over the past decade, yeast models with powerful genetics and biochemical techniques have contributed to uncovering the mechanism of the co-translational quality control system, which eliminates the harmful products generated from aberrant translation. We here summarize the current knowledge of the molecular mechanism of the co-translational quality control systems in yeast, which eliminate the incomplete nascent chain, improper mRNAs, and faulty ribosomes to maintain cellular protein homeostasis.

## 1. Introduction

Cells continually monitor the accuracy of gene expression processes and eliminate aberrant and harmful gene products. In the translation cycle, ribosome behavior is key to judging the quality of mRNA and nascent proteins [1,2,3]. Many potential problems, including genetic mutations, mRNA processing errors, insufficiency of charged tRNAs, and a faulty ribosome result in abnormal behavior of ribosomes, such as stalling during translation [4,5]. Quality control sensor proteins recognize the aberrant behavior of ribosomes and induce the downstream pathways of ribosome rescue, mRNA decay, nascent protein degradation, and stress responses [1,2,3,6,7,8,9,10,11]. Rapid recognition and resolution of problematic events in the translation are critical for maintaining cellular protein homeostasis. Defects in quality control systems lead to the accumulation of protein aggregation, which is associated with various diseases, such as neurodegeneration [9,12,13,14,15,16]. Over the past decade, numerous quality control pathways coupled with translation have been discovered using yeast systems. Interestingly, most of these pathways are well conserved from yeast to mammals. In this review article, we summarize the current knowledge of the molecular mechanism of co-translational quality control uncovered by yeast models.

## 2. Ribosome Collision as a Common Feature of Translational Stress

mRNA translation is tightly coupled with protein maturation processes, such as protein folding [17,18,19,20,21], protein targeting [22,23,24], protein assembly [25,26,27], and mRNA decay [28,29,30], and thus the translational speed of individual mRNA is highly coordinated with associated biological processes by synonymous codon usage to produce fully functional mature proteins. Accordingly, the proper translation speeds range widely in the cellular translation pool. Thus, it had been unclear for a long time how aberrant and harmful ribosome stalling can be determined from the diverse cellular translation pool by the cell to initiate downstream quality control pathways. Recent studies uncover that the selectivity for aberrant and harmful translation arrest is determined by ribosome queuing rather than simply by translation slowdown to avoid targeting the pausing of programmed ribosome stalling [31,32,33]. This mechanism can identify many types of aberrant ribosome stalling in diverse translational pools. Cryo-electron microscopy (Cryo-EM) analysis of colliding ribosomes reveals that ribosome queuing is not merely juxtaposed a ribosome on the same mRNA; rather, this provides the unique structural architecture at the collision interface [31,32,34], which can serve as a scaffold for the sensor proteins to induce the downstream pathways (Figure 1). 

In yeast, one of the sensor proteins Hel2 recognizes ribosome collision as a problematic event in translation and induces the downstream quality control pathways to degrade incomplete nascent chain and aberrant mRNA (Figure 1), which are called ribosome-associated quality control (RQC) and no-go mRNA decay (NGD), respectively [32,33,34,35,36]. On the colliding ribosomes, Hel2 ubiquitinates ribosomal protein uS10 to initiate non-canonical subunit dissociation of lead stalling ribosomes, which allows for resuming translation by following ribosomes on the same mRNA [34]. The Hel2-mediated pathway for clearing the ribosome collision is perfectly conserved in mammalian cells. ZNF598 (mammalian Hel2 homolog) has same function in mammalian cells and resolves the ribosome queuing [31,37,38,39,40,41]. Both models are reconstituted by the in vitro translation system using yeast or rabbit reticulocyte lysate [34,42]. 

In parallel with the Hel2-mediated pathway, the integrated stress response (ISR) is activated by the ribosome collision. The GCN complex, composed of Gcn1 and Gcn20, directly binds to the colliding ribosomes and recruits Gcn2 via interaction with Gcn1 [43]. Activation of Gcn2 results in the phosphorylation of eIF2α and inhibits the further round of translation initiation [44], alleviating the formation of the ribosome traffic jams (Figure 1). Furthermore, the lack of Hel2 leads to the activation of the eIF2α phosphorylation [45], suggesting that ISR compensates for the Hel2-mediated quality control pathway to alleviate the accumulation of ribosome collisions when the availability of Hel2 is limited, such as globally severe ribosome stalling. 

Although ISR and Hel2-mediated pathways are well conserved from yeast to mammal, an additional cellular response, the ribotoxic stress response (RSR), is activated by a severe ribosome collision in the mammalian cell [46,47,48,49]. The widespread ribosome collisions, which cannot be solved by only the ZNF598-mediated quality control pathway, are recognized by another sensor protein ZAKα (mitogen-activated protein kinase kinase kinase: MAPKKK) and triggers RSR to regulate cell fate decisions (Figure 1) [46,47,48,49]. The GCN complex phosphorylates eIF2α to activate ISR in response to less severe collisions; however, if the ribosome collisions are more severe, ZAKα activates the RSR pathway by inducing phosphorylation of p38 or c-Jun n-terminal kinase, which can lead to cell cycle arrest or apoptosis, respectively [46,47,48,49].

## 3. The Clearance System of Ribosome Traffic Jam

Ribosome traffic jams are formed by various types of translational arrest; the most common case is the poly(A) tract generated by premature polyadenylation [6,50,51,52,53,54], which is estimated to be 1–5% of intercellular mRNAs [55,56]. Poly arginine stretches including CGA rare codons have been routinely used as reporters of ribosome stalling in yeast [36,52]. Recently, such a polybasic stretch was found in the endogenous *SDD1* mRNA and induces the Hel2-mediated quality control pathway [34]. The ribosome profiling clearly shows the several peaks upstream of the pause site with a periodicity of roughly 10 codons, indicating that translational arrest on the *SDD1* mRNA results in the collision of the following ribosomes with the first stalled one [34]. The ribosome collision on the model mRNA can be reconstituted by the in vitro translation system using yeast lysate; Hel2 specifically ubiquitinates ribosomal protein uS10 in more than disome structures [32,34]. Furthermore, its structural analysis by cryo-EM analysis reveals that the collision interface creates unique structural architectures, which is a 40S head-to-head contact between RACK1 of the leading ribosome and RACK1 of the colliding ribosome [32,34]. Notably, RACK1 is well known as a scaffold protein in the ribosome [57] and is located adjacent to the ubiquitination sites by Hel2 and required for RQC and NGD [58], suggesting that the RACK1-RACK1 site in the colliding ribosome could serve as a scaffold for Hel2 to induce the downstream pathways. However, the structure of Hel2-associated ribosomes is yet to be determined. 

Hel2 can elongate the K63-linked polyubiquitin chain on the uS10 (Figure 2) [32,59]. This ubiquitin chain functions as the marker of aberrant translation and is recognized by the ribosome quality control triggering (RQT) complex (Figure 2), which is composed of RNA helicase Slh1, ubiquitin-binding protein Cue3, and zinc-finger protein Rqt4 [34,36]. Although Cue3 has been considered the sole responsible factor for the recognition of uS10 ubiquitination, a recent study uncovers that not only Cue3, but also Rqt4, can interact with the K63-linked ubiquitin chain and facilitate the recruitment of the RQT complex into the colliding ribosomes (Figure 2) [59]. After the recruitment, the RQT complex specifically disassembles the leading ribosome, which enables the following ribosomes to resume translation (Figure 2); the splitting reaction by the RQT complex can be reconstituted by the in vitro translation system using yeast lysate [34,59]. Additionally, the recent cryo-EM analysis visualizes that the RQT complex engages the 40S subunit of the lead ribosome and can switch between two conformations. This result proposes the model in which the Slh1 helicase subunit of the RQT complex applies a pulling force on the mRNA, causing destabilizing conformational changes in the 40S subunit [60]. The clearance system of ribosome traffic jams described here seems to be well conserved from yeast to mammals [31,38,40,41,42,61].

## 4. Degradation Pathway of Incomplete Nascent Chains and Aberrant mRNAs

Hel2-mediated non-canonical ribosome dissociation always produces an incomplete nascent chain, which has potentially toxic properties [7,9,12]. Thus, it is immediately degraded by the surveillance system (Figure 3). After ribosome dissociation by the RQT complex, the incomplete nascent chain is still retained on the dissociated 60S subunit [62]. The 60S subunit containing an incomplete nascent chain is sensed as an abnormal 60S subunit by the ribosome quality control (RQC) factor Rqc2, followed by the recruitment of E3 ubiquitin ligase Ltn1 [62,63,64,65] (Figure 3). Ltn1 ubiquitinates the nascent chain with a K48-linked ubiquitin chain, which further recruits an AAA-ATPase Cdc48 [62,66,67]. Cdc48 extracts the nascent chain from the ribosome tunnel and delivers it to the proteasome for degradation [66,67] (Figure 3). Before the extraction step by Cdc48, the nascent chain trapped with the 60S subunit must be released from the tRNA, which is covalently linked. In the canonical translation termination, the release factor eRF1 catalyzes the hydrolysis of the peptidyl-tRNA to release the nascent peptide; however, the release of the nascent peptide from the RQC complex is carried out by the cleavage of tRNA by Vms1 instead of hydrolysis [68] (Figure 3). Vms1 binds to the A-site of the 60S subunit together with ABCF-type ATPase Arb1 in the E-site, which stabilizes the delocalized A73 of peptidyl-tRNA and stimulates Vms1-dependent tRNA cleavage [68].

Aside from the recruitment of Ltn1, Rqc2 is responsible for the CATylation, a c-terminal alanine and threonine extension reaction that is independent of mRNA and the 40S subunit [7,12,68,69,70]. When the nascent polypeptide is exposed from the ribosome exit, the tunnel has no lysine residues accessible for ubiquitination by Ltn1. In this case, Rqc2 recruits charge tRNAs independently of mRNA to extend the carboxy terminus of the nascent chain with alanine and threonine residues (called CAT tails) [69,70], which pushes the polypeptide out of the ribosomal tunnel until a lysine residue is exposed, which is targeted by Ltn1 [69] (Figure 3). The CATylation helps the Ltn1-dependent degradation of the incomplete nascent peptide, but also induces protein aggregation in the absence of Ltn1, which leads to the imbalance of cellular protein homeostasis [71,72]. The RQC pathway for degradation of the incomplete nascent chain is highly conserved from yeast to mammals [73,74,75,76]. In addition, recent studies reveal the conservation of the RQC pathway in not only eukaryotes, but also prokaryotes [77,78].

Hel2-associated ribosome profiling reveals that Hel2 binds preferentially to the pre-engaged secretory ribosome-nascent chain complexes (RNCs) lacking signal recognition particles (SRP), which translates upstream of targeting signals [35]. Moreover, the mitochondrial defects caused by insufficient SRPs are enhanced by Hel2 deletion, along with the mistargeting of secretory proteins into mitochondria [35]. This observation suggests the competition between Hel2 and SRP in the early phase of translation to determine the fate of secretory protein translating ribosomes. The Hel2 mediated-quality control pathway could function as preventive quality control in the secretory pathway. In addition, recent studies reveal that eIF3, a multi-subunit complex for translation initiation, is still associated with translating ribosomes in the elongation step after initiation [79,80,81], suggesting the involvement of eIF3 in the events in the early phase of translation such as the SRP-mediated targeting pathway. The proteomic analysis of the eIF3 interactome suggests that eIF3 serves to recruit protein quality-control factors to ribosomes [81,82]. Furthermore, loss of eIF3’s elongation and subcellular targeting activity leads to mitochondrial and skeletal muscle dysfunction [81]. Thus, eIF3 could be involved in co-translational protein degradation.

The collision sensor Hel2 induces not only peptide degradation but also mRNA decay, which is called no-go decay (NGD) [32,33,36]. In the NGD pathway, the mRNA is endonucleolytic cleaved by ubiquitin-binding nuclease Cue2 in the vicinity of the stalling ribosome [83], resulting in the rapid degradation of the cleaved mRNAs by the exoribonuclease Xrn1 and Ski complex [84,85,86,87,88,89]. The Cue2-mediated endonucleolytic cleavage depends on the ribosome ubiquitination by the Hel2 [83]. The Hel2 can elongate the K63-linked ubiquitin chain on the two different ribosomal proteins, uS10 and eS7 [32]. The ubiquitination of uS10 is essential for the ribosome dissociation by the RQT complex to induce the RQC pathway so that uS10-dependent NGD could be coupled with RQT-mediated ribosome dissociation, which is called NGD^RQC+^ [32]. Indeed, the uS10 ubiquitination-dependent mRNA cleavage is observed within the disome unit [32]; since the cleavage site is covered within the colliding ribosome, the ribosome splitting activity by the RQT complex seems to be necessary to induce this reaction. On the other hand, if the ribosome dissociation by the RQT complex is blocked, mRNA is cleaved at the further upstream region of the leading disome unit, which is called NGD^RQC–^ [32]. The NGD^RQC–^ depends on the ubiquitination of eS7, but not uS10. eS7 is initially mono-ubiquitinated by the Not4 E3 ubiquitin ligase within the Ccr4-NOT complex [28,32,90], with subsequent K63-linked ubiquitin chain elongated by Hel2 [32]. Furthermore, mRNA cleavage in the NGD^RQC–^ takes place at the edge of the ribosome between the colliding ones, which is not required in ribosome dissociation. Thus, this mechanism could function as a reserve system for the NGD^RQC+^.

Cue2 is the primary endonuclease-initiating NGD that cleaves mRNAs in the A site of collided ribosomes in presence of Slh1 [83]. Cue2 contains two CUE domains and one UBA domain, and the ubiquitin-binding activity of Cue2 is required for NGD^RQC–^ but not for NGD^RQC+^, and it involves the first two n-terminal Cue domains [91]. These indicate that two modes of Cue2-mediated mRNA cleavage with distinct substrate recognition initiate no-go decay.

## 5. Non-Functional Ribosome Decay

The defects in the ribosome could also lead to translational arrest, which is a similar situation to defective mRNA. In this case, faulty ribosomes are degraded through a process termed nonfunctional rRNA decay (NRD) [92,93,94,95]. The functional defects in the small (40S) or large (60S) subunit are discriminated and sorted into the different pathways, 18S or 25S NRD, respectively [94,95]. 18S NRD is initiated by monoubiquitination of uS3 by the E3 ligase Mag2, followed by K63 polyubiquitination by additional E3 ligase Hel2, Rsp5, or Fap1 [96,97]. After ubiquitination, the ribosome is disassembled in the RQT complex- or Dom34-dependent manner [96,97]; afterward, only the defective 40S subunit could be degraded. 

The most famous substrate for 18S NRD is the non-functional 40S subunit with the mutation in the decoding center, which is used as the model substrate for the analysis of 18S NRD [92,93] (Figure 4). The selective ribosome profiling in the recent report reveals that the decoding-defective ribosomes are mainly stalled at the initiation codon together with the E3 ligase Fap1 [96], which elongates the K63-linked ubiquitin chain on the uS3 after the initial monoubiquitination by Mag2 [96] (Figure 4). In contrast to Fap1, the initial sensor Mag2 distributes the whole open reading frame (ORF) and preferentially binds to the slow-translating ribosome on the non-optimal codon [96] (Figure 4). These observations indicate that 18S NRD initiates after the double-layer judgments; Mag2 adds the monoubiquitination on the uS3 into the slow-translating ribosomes (first selection), but it could not be enough to induce 18S NRD. If the defective ribosome with monoubiquitinated uS3, marked by Mag2, is recognized by Fap1 (2nd selection), the K63-linked ubiquitin chain can be elongated on the uS3, initiating the 18S NRD pathway (Figure 4).

Recent cryo-EM analysis revealed that Fap1 notably binds to the 40S subunit and mRNA on both exit and entry sites [96], suggesting that Fap1 cannot bind to the colliding ribosome because the Fap1 binding site clashes the colliding interface. Thus, this indicates that Fap1 acts as the monosome-stalling sensor, which can detect aberrant translational arrest without ribosome collision.

## 6. Future Perspectives

Over the past five years, many new mechanisms underlying the quality control associated with ribosome collision have been discovered using the yeast model system, which is well-conserved from yeast to human. However, it remains to be understood how different pathways induced by ribosome collision are selected and organized by the cell. For the quality control pathway, the ubiquitination of ribosomes is the key event in activating the downstream pathway. Indeed, the diverse ubiquitinations of the ribosome are observed in the cell, e.g., the ubiquitination of ribosomal proteins uS10, uS3, eS7, and uL23 are essential to initiate different pathways (the RQC pathway [32,34,36], the 18S non-functional rRNA decay (18S NRD) [96,97], the no-go decay (NGD) [32], and the ribophagy [98], respectively). However, the decoding mechanism of the ribosome ubiquitin code remains largely unknown. Thus, the elucidation of the decoding mechanism of ribosome ubiquitin code is one of the future directions to uncover more details of the molecular mechanism. 

Recent findings let us consider that the ribosome collision seems to be a key signal to connect to environmental conditions, which could be a sensor of the cellular state. For example, ZAKα senses ribosome collision and determines cell fate: cell cycle arrest or apoptosis [46,48,99]. Furthermore, recent studies report that ribosomal collisions have been implicated in the coactivation of the innate immune response through cyclic GMP-AMP synthase (cGAS) [100] and that the degradation of the nascent chain by the RQC pathway contributes to efficient MHC-I presentation [101]. Thus, the ribosome behaviors including collision seem to be a good indicator to monitor the global translational status in response to environmental changes. The cell could select the appropriate unknown pathways to adapt to the environmental stresses by sensing the translational conditions. In future studies, a deeper understanding of how cells deal with ribosome behaviors will open new frontiers in understanding how cells respond to environmental stresses.

## Figures and Tables

**Figure 1 biomolecules-13-00317-f001:**
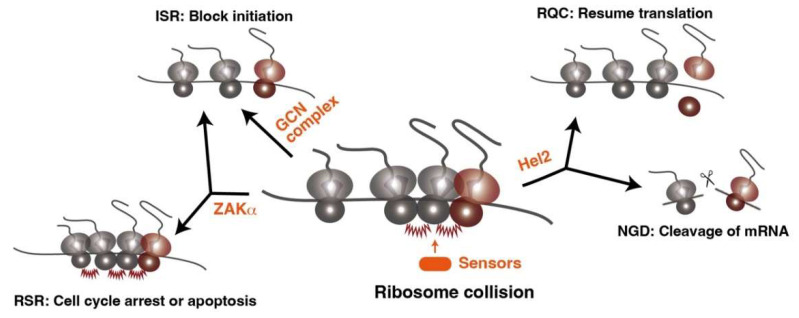
The response to the ribosome collision. Several sensor proteins detect the ribosome collision and induce different pathways. Hel2 (Znf598 in mammals) induces ribosome-associated quality control (RQC) and the no-go decay (NGD). The general control nonderepressible (GCN) complex induces an integrated stress response (ISR). ZAKα induces cell cycle arrest or apoptosis via the p38 or c-Jun pathway, respectively, which are called ribotoxic stress response (RSR).

**Figure 2 biomolecules-13-00317-f002:**
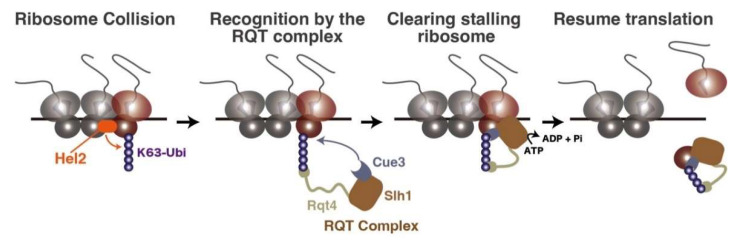
Ribosome quality control triggering step. The ribosome collision is recognized by E3 ligase Hel2, which elongates the K63-linked poly-ubiquitination on the ribosomal protein uS10. The K63-linked polyubiquitination is recognized by the RQT complex via its components Cue3 and Rqt4, inducing the non-canonical ribosome dissociation for the leading stalling ribosome to resume translation.

**Figure 3 biomolecules-13-00317-f003:**
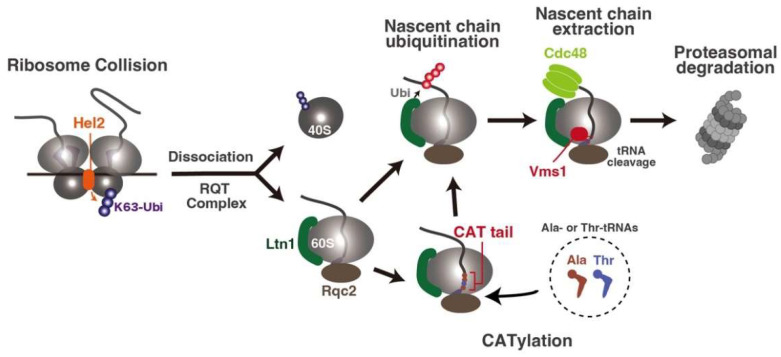
Degradation pathway of incomplete nascent chains. After the Hel2-mediated ribosome splitting, the incomplete nascent chain on the dissociated 60S subunit is recognized by Rqc2 and E3 ligase Ltn1 and is ubiquitinated. If Ltn1 cannot ubiquitinate the nascent chain, e.g., no lysine residue within the Ltn1-accessible region, Rqt2 recruits the alanine- (Ala-) or threonine- (Thr-) charged tRNAs to extend the c-terminal of the nascent chain (CATylation). It contributes to pushing out the lysine residue within the ribosome tunnel. After ubiquitination of the nascent chain, peptidyl-tRNA is cleaved by Vms1 at the cytosine-cytosine-adenine (CCA) end of tRNA; afterward, the nascent chain is extracted from the ribosome tunnel by Cdc48 and degraded by the proteasome.

**Figure 4 biomolecules-13-00317-f004:**
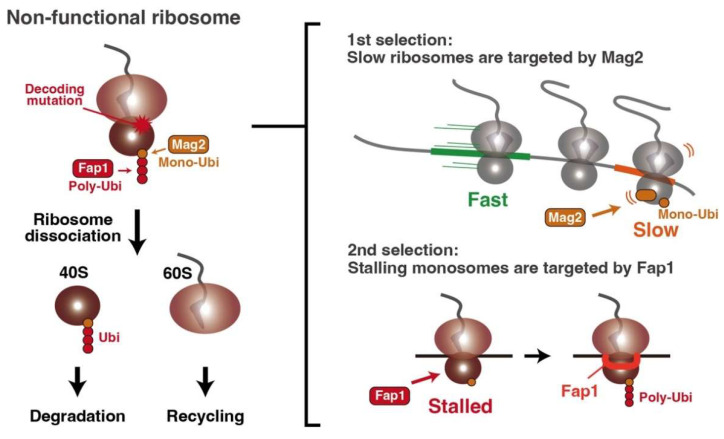
Degradation pathway for the non-functional ribosome. The decoding-defective ribosomes are initially monoubiquitinated by Mag2 and then polyubiquitinated by Fap1 to initiate 18S nonfunctional rRNA decay (selective degradation of small subunit). In the first selection, Mag2 recognizes the slow translating ribosome and monoubiquitinates ribosomal protein uS3. After the first selection step, Fap1 recognizes stalling monoribosomes and elongates the K63-linked polyubiquitin chain on the Mag2-mediated monoubiquitinated uS3. This tandem ubiquitin system triggers the non-functional 18S rRNA decay.

## Data Availability

Not applicable.

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
