# Peer review of "Co-Translational Quality Control Induced by Translational Arrest"

_biomolecules, 2023, doi:10.3390/biom13020317_

Round 1

Reviewer 1 Report

The review manuscript “Co-translational quality control induced by translational arrest” by Yoshitaka Matsuo and Toshifumi Inada; focuses on ribosome quality control pathways, reviewing the emerging quality controls pathways in eukaryotic cells, with an emphasis on ribosome collisions and non-functional ribosome decay.

The review is well written, the figures are highly clear and the manuscript encompasses the recent advances in this rapidly evolving topic. The manuscript is an important contribution to the field of protein biogenesis providing a comprehensive discussion on the growing number of factors involve in RQC, their mechanism of action, including future prospectives on entropic pulling as well as ribosome collisions as a key signal in the cellular various stress responses.

Minor changes are required:

·       In segment 4, degradation pathways on incomplete nascent chains and aberrant mRNAs, the authors should discuss conservation of RQC in prokaryotes, specifically CAT tails mimicry (see: Filbeck, S., Cerullo, F., Paternoga, H., Tsaprailis, G., Joazeiro, C.* and Pfeffer S.* 2021. Mimicry of Canonical Translation Elongation Underlies Alanine Tail Synthesis in RQC. Mol. Cell 81:104-114 Mol. Cell 2021, 81:6-7).

·       The authors should elaborate on the possible involvement of eIF3 in co-translational protein degradation, as eIF3 interactome revealed eIF3, via its eIF3e subunit, serves to recruit protein quality-control factors to ribosomes, including the proteasome, with impact on mitochondrial homeostasis, and muscle health (Sha, Z.,…, Glickman, M.H, & Wolf, D.A., 2009; Lin Y.Y.,…,Wolf, D.A., 2020)

·       Reference 59 was published today, thus it can be updated (Matsuo, Y., Uchihashi, T. & Inada, T. Decoding of the ubiquitin code for clearance of colliding ribosomes by the RQT complex. Nat Commun 14, 79 (2023).

·       In future prospective segment, the authors discuss colliding ribosomes implication in the innate immune response. The authors should also discuss the contribution of the Listerin ubiquitin ligase to MHC-I peptide repertoire in human cells (Trentini, D.B., Pecoraro, M., Tiwary, S., Cox, J., Mann, M., Hipp, M.S., et al. (2020). Role for ribosome-associated quality control in sampling proteins for MHC class I-mediated antigen presentation. Proc Natl Acad Sci U S A 117(8), 4099-4108).
In the study, mass spectrometry based immunopeptidome analysis of WT and Listerin-KO cells revealed how Listerin deletion affects the degradation of endogenous human proteins, their presentation and the innate immune response (Trentini et al., 2020).

Author Response

We thank reviewer for the helpful, and insightful comments. In our detailed response, the reviewers’ comments are Italicized whereas our response is in Roman typeface with blue color.

The review manuscript “Co-translational quality control induced by translational arrest” by Yoshitaka Matsuo and Toshifumi Inada; focuses on ribosome quality control pathways, reviewing the emerging quality controls pathways in eukaryotic cells, with an emphasis on ribosome collisions and non-functional ribosome decay. 

The review is well written, the figures are highly clear and the manuscript encompasses the recent advances in this rapidly evolving topic. The manuscript is an important contribution to the field of protein biogenesis providing a comprehensive discussion on the growing number of factors involve in RQC, their mechanism of action, including future prospectives on entropic pulling as well as ribosome collisions as a key signal in the cellular various stress responses. 

Minor changes are required:

  • In segment 4, degradation pathways on incomplete nascent chains and aberrant mRNAs, the authors should discuss conservation of RQC in prokaryotes, specifically CAT tails mimicry (see: Filbeck, S., Cerullo, F., Paternoga, H., Tsaprailis, G., Joazeiro, C.* and Pfeffer S.* 2021. Mimicry of Canonical Translation Elongation Underlies Alanine Tail Synthesis in RQC. Mol. Cell 81:104-114 Mol. Cell 2021, 81:6-7).

We have added the following sentence and the references at the end of segment 4.

“In addition, recent studies reveal the conservation of the RQC pathway in not only eukaryotes but also prokaryotes [76, 77]”.

Ref. 76 and 77 were added in the reference list.

  1. Lytvynenko, I.; Paternoga, H.; Thrun, A.; Balke, A.; Müller, T.A.; Chiang, C.H.; Nagler, K.; Tsaprailis, G.; Anders, S.; Bischofs, I.; et al. Alanine Tails Signal Proteolysis in Bacterial Ribosome-Associated Quality Control. Cell 2019, 178, 76-90.e22, doi:10.1016/j.cell.2019.05.002.
  2. Filbeck, S.; Cerullo, F.; Paternoga, H.; Tsaprailis, G.; Joazeiro, C.A.P.; Pfeffer, S. Mimicry of Canonical Translation Elongation Underlies Alanine Tail Synthesis in RQC. Mol Cell 2021, 81, 104-114.e106, doi:10.1016/j.molcel.2020.11.001.

  • The authors should elaborate on the possible involvement of eIF3 in co-translational protein degradation, as eIF3 interactome revealed eIF3, via its eIF3e subunit, serves to recruit protein quality-control factors to ribosomes, including the proteasome, with impact on mitochondrial homeostasis, and muscle health (Sha, Z.,…, Glickman, M.H, & Wolf, D.A., 2009; Lin Y.Y.,…,Wolf, D.A., 2020)

      As suggested reviewer, we have added the following sentence and references in segment 4.

      Hel2-associated ribosome profiling reveals that Hel2 binds preferentially to the pre-engaged secretory ribosome-nascent chain complexes (RNCs) lacking signal recognition particle (SRP), which translates upstream of targeting signals [35]. Moreover, the mitochondrial defects caused by insufficient SRP are enhanced by hel2 deletion, along with the mistargeting of secretory proteins into mitochondria [35]. This observation suggests the competition between Hel2 and SRP in the early phase of translation to determine the fate of secretory protein translating ribosomes; the Hel2 mediated-quality control pathway could function as preventive quality control in the secretory pathway. In addition, recent studies reveal that eIF3, a multi-subunit complex for translation initiation, is still associated with translating ribosomes in the elongation step after initiation [78-80], suggesting the involvement of eIF3 in the events in the early phase of translation such as the SRP-mediated targeting pathway. The proteomic analysis of the eIF3 interactome suggests that eIF3 serves to recruit protein quality-control factors to ribosomes [80,81]. Furthermore, loss of eIF3’s elongation and subcellular targeting activity leads to mitochondrial and skeletal muscle dysfunction [80]. Thus, eIF3 could be involved in co-translational protein degradation.

Ref. 78-81 were added in the reference list

  1. Wagner, S.; Herrmannová, A.; Hronová, V.; Gunišová, S.; Sen, N.D.; Hannan, R.D.; Hinnebusch, A.G.; Shirokikh, N.E.; Preiss, T.; Valášek, L.S. Selective Translation Complex Profiling Reveals Staged Initiation and Co-translational Assembly of Initiation Factor Complexes. Mol Cell 2020, 79, 546-560.e547, doi:10.1016/j.molcel.2020.06.004.
  2. Bohlen, J.; Fenzl, K.; Kramer, G.; Bukau, B.; Teleman, A.A. Selective 40S Footprinting Reveals Cap-Tethered Ribosome Scanning in Human Cells. Mol Cell 2020, 79, 561-574.e565, doi:10.1016/j.molcel.2020.06.005.
  3. Lin, Y.; Li, F.; Huang, L.; Polte, C.; Duan, H.; Fang, J.; Sun, L.; Xing, X.; Tian, G.; Cheng, Y.; et al. eIF3 Associates with 80S Ribosomes to Promote Translation Elongation, Mitochondrial Homeostasis, and Muscle Health. Mol Cell 2020, 79, 575-587 e577, doi:10.1016/j.molcel.2020.06.003.
  4. Sha, Z.; Brill, L.M.; Cabrera, R.; Kleifeld, O.; Scheliga, J.S.; Glickman, M.H.; Chang, E.C.; Wolf, D.A. The eIF3 interactome reveals the translasome, a supercomplex linking protein synthesis and degradation machineries. Mol Cell 2009, 36, 141-152, doi:10.1016/j.molcel.2009.09.026.
  • Reference 59 was published today, thus it can be updated (Matsuo, Y., Uchihashi, T. & Inada, T. Decoding of the ubiquitin code for clearance of colliding ribosomes by the RQT complex. Nat Commun 14, 79 (2023).

We have corrected it.

  • In future prospective segment, the authors discuss colliding ribosomes implication in the innate immune response. The authors should also discuss the contribution of the Listerin ubiquitin ligase to MHC-I peptide repertoire in human cells (Trentini, D.B., Pecoraro, M., Tiwary, S., Cox, J., Mann, M., Hipp, M.S., et al. (2020). Role for ribosome-associated quality control in sampling proteins for MHC class I-mediated antigen presentation. Proc Natl Acad Sci U S A 117(8), 4099-4108). 
    In the study, mass spectrometry based immunopeptidome analysis of WT and Listerin-KO cells revealed how Listerin deletion affects the degradation of endogenous human proteins, their presentation and the innate immune response (Trentini et al., 2020).

      As suggested reviewer, we have added the following sentence and references in perspective section.

“and that the degradation of the nascent chain by RQC pathway contributes to efficient MHC-I presentation [100].”

Ref. 100 was added in the reference list.

  1. Trentini, D.B.; Pecoraro, M.; Tiwary, S.; Cox, J.; Mann, M.; Hipp, M.S.; Hartl, F.U. Role for ribosome-associated quality control in sampling proteins for MHC class I-mediated antigen presentation. Proc Natl Acad Sci U S A 2020, doi:10.1073/pnas.1914401117.

Reviewer 2 Report

In this manuscript, the authors reviewed the current knowledge of the molecular mechanism of the co-translational quality control systems in yeast, with a particular focus on ribosome collision. This well-structured review manuscript should attract readers in the related filed with detailed information on ribosome collision response, leading stalling ribosome dissociation for translation resuming, and the degradation pathway of incomplete nascent chains. This reviewer only has one suggestion for  authors to address in prior to publication.

This review has the feeling that according to the manuscript, all the molecular mechanisms of the co-translational quality control of ribosome collision are clear and uncovered, which is not the case. The authors are encouraged to discuss the aspects about what need to be further elucidated and give more details regarding the future perspectives in the field.

Author Response

We thank reviewer for the helpful, and insightful comments. In our detailed response, the reviewers’ comments are Italicized whereas our response is in Roman typeface with blue color.

In this manuscript, the authors reviewed the current knowledge of the molecular mechanism of the co-translational quality control systems in yeast, with a particular focus on ribosome collision. This well-structured review manuscript should attract readers in the related filed with detailed information on ribosome collision response, leading stalling ribosome dissociation for translation resuming, and the degradation pathway of incomplete nascent chains. This reviewer only has one suggestion for authors to address in prior to publication.

This review has the feeling that according to the manuscript, all the molecular mechanisms of the co-translational quality control of ribosome collision are clear and uncovered, which is not the case. The authors are encouraged to discuss the aspects about what need to be further elucidated and give more details regarding the future perspectives in the field.

As suggested by the reviewer, we have added the sentence as follows in the 1st paragraph of the perspective session, which discussed the future direction for uncovering the molecular mechanism of co-translational quality control.

However, it remains to be understood how different pathways induced by ribosome collision are selected and organized by the cell. For the quality control pathway, the ubiquitination of ribosomes is the key event to activating the downstream pathway. Indeed, the diverse ubiquitinations of the ribosome are observed in the cell; e.g., the ubiquitination of ribosomal proteins uS10, uS3, eS7, and uL23 are essential to initiate different pathways: the RQC pathway [32,34,36], the 18 S non-functional rRNA decay (18 S NRD) [95,96], the no-go decay (NGD) [32], and the ribophagy [97], respectively. However, the decoding mechanism of ribosome ubiquitin-code remains largely unknown. Thus, the elucidation of the decoding mechanism of ribosome ubiquitin-code is one of the future directions to uncover more details of the molecular mechanism.

Ref. 97 was added in the reference list

  1. Ossareh-Nazari, B.; Nino, C.A.; Bengtson, M.H.; Lee, J.W.; Joazeiro, C.A.; Dargemont, C. Ubiquitylation by the Ltn1 E3 ligase protects 60S ribosomes from starvation-induced selective autophagy. J Cell Biol 2014, doi:10.1083/jcb.201308139.